# Clinical and Mutation Spectrum of Autosomal Recessive Non-Syndromic Oculocutaneous Albinism (nsOCA) in Pakistan: A Review

**DOI:** 10.3390/genes13061072

**Published:** 2022-06-16

**Authors:** Muhammad Ikram Ullah

**Affiliations:** Department of Clinical Laboratory Sciences, College of Applied Medical Sciences, Jouf University, Sakaka 75471, Aljouf, Saudi Arabia; mikramullah@ju.edu.sa; Tel.: +966-532-783-173

**Keywords:** albinism, consanguinity, clinical spectrum, genetic mutations, non-syndromic oculocutaneous albinism

## Abstract

Oculocutaneous albinism (OCA) is an autosomal recessive syndromic and non-syndromic defect with deficient or a complete lack of the melanin pigment. The characteristics of OCA appears in skin, hair, and eyes with variable degree of pigmentation. Clinical manifestations of OCA include nystagmus, photophobia, reduced visual acuity, hypo-plastic macula, and iris trans-illumination. There are eight OCA types (OCA1–8) documented with non-syndromic characteristics. Molecular studies identified seven genes linked to the OCA phenotype (*TYR*, *OCA2*, *TYRP1*, *SLC45A2*, *SLC24A5*, *C10orf11,* and *DCT*) and one locus (OCA5) in consanguineous and sporadic albinism. The complications of OCA result in skin cancer and variable syndromes such as Hermansky–Pudlak syndrome (HPS) Chediak–Higashi syndrome (CHS). In the Pakistani population, autosomal recessive non-syndromic OCA is common and is associated with a large number of consanguineous families, and mutations in genes of non-syndromic types are reported. This review highlights the updates on the genetic mutation of OCA genes reported from Pakistani families. Several studies reported the genetic mutations in OCA1, OCA2, OCA3, OCA4, and OCA6 albinism in Pakistani families. A locus, OCA5, was also reported from the Pakistani population, but the gene has not been identified. A new type of OCA8 was identified due to the *DCT* gene mutation, and it is also reviewed here.

## 1. Introduction

Albinism is a complex group of disorders that lead to a reduced production or complete lack of pigment–protein melanin, resulting in a characteristic white or albino appearance of hair, skin, and eyes. There are two forms of albinism, categorized as ocular (only pertaining to eyes) and oculocutaneous albinism (OCA associated with eyes, hair, and skin). In these classes, the most commonly distributed entity is OCA that is reported from different ethnic, language, and ancestral groups worldwide. Clinical characteristics of OCA are associated with dramatic effects on skin, hair, and eyes. Clinical presentation of skin is widely diverse and linked with a degree of severity contingent on OCA sub-clinical types. The ocular arrangements depend on the signaling for melanin during the uterine development of the eye; hence, diverted optic nerve fibers develop ocular manifestations of the related defects [1,2,3].

At present, there are eight clinical types of non-syndromic oculocutaneous albinism (OCA1–8) that are linked to variable clinical features pertaining to skin, hair, and eyes. In non-syndromic conditions, each clinical type is linked to a specific causative gene. On the other hand, syndromic albinism is comprised of a group of signs and symptoms including nystagmus, visual acuity, foveal hypoplasia, and bleeding defects. At present, there are different classes of syndromic albinism in which genetic mutations are documented. There is extensive literature that describes the syndromic forms of albinism including Hermansky–Pudlak syndrome (HPS) and Chediak–Higashi syndrome (CHS). Among syndromic forms, HPS is the most common variant, and eleven subtypes (HPS1–11) are documented. The lack of ocular and dermal pigment in syndromic types of albinism is comparable to non-syndromic albinism. However, the genes impede syndromic forms that code the proteins that are extensively involved in cellular function. In syndromic albinism, the loss of function mutations produces predictable consequences. In HPS, the bleeding diathesis causes variation in genes that encode lysosomal synthesis (not only melanin synthesis involvement) and infection progression in CHS. There are other anomalies that present similar to albinism including congenital nystagmus and generalized hypopigmentation. The examples of syndromes associated with similar mutated genes as in the OCA2 type are Angelman (AS) and Prader–Willi (PWS) syndromes [4,5,6].

## 2. Literature Review

### 2.1. OCA Albinism

OCA is a genetic disorder with an autosomal recessive mode of inheritance in which there is a complete absence or decreased production of melanin that causes hypo-pigmentation of hair, skin, and eyes. Each parent transfers a copy of the faulty gene to the affected child in OCA. Physically normal carriers of OCA can transfer the disease to the next generation due to the presence of only one copy of the defective gene or compound heterozygous mutations in two different alleles of OCA genes, which have also been identified in different populations of the world, establishing the heterogeneity of this disorder [7].

Albinism is also present in syndromic form with variable signs and symptoms such as neurological problems, with high susceptible to infections [8]. Hermansky–Pudlak, Chediak–Higashi, Griscelli Syndrome, Prader–Willi, Angelman, Cross–McKusick–Breen, Griscelli, Elejalde, and Waardenburg Syndrome type II (WS2) are the main examples of syndromic albinism. All of these syndromes except WS2, are autosomal recessive and can be diagnosed by clinical, biochemical, and genetic testing.

On the basis of genetic testing, non-syndromic OCA is further divided into eight subtypes (OCA1–8), among which OCA1 is the most common that affects almost 50% of albino individuals worldwide [9].

### 2.2. Melanin Biosynthesis Pathway and Its Link to OCA Types

In melanosomes, the molecular mechanism regulates the synthesis of melanin pigment by melanocyte cells. In this pathway, there are a number of genes and contributing enzymes that take part in multiple reactions to make melanin products. In a chemical reaction, L-tyrosine is converted into a 3,4-dihydroxyphenylalanine (DOPA) intermediate that then results in the formation of dopaquinone. From dopaquinone, the melanin synthetic pathway splits into two pathways leading to eumelanin formation (brownish to black coloration) and pheomelanin (reddish to yellow pigmentation) ultimately (Figure 1). Previous studies documented the crucial role of the genes (*TYR*, *OCA2*, *TYRP1*, *SLC45A2*, *SLC24A5*, *C10orf11*, and *DCT*) in modulating the regulation of melanin production [10,11]. The initiation of the pathway is regulated by the tyrosinase enzyme that is a membrane-bound copper-containing enzyme that causes the hydroxylation of l-tyrosine into L-DOPA, and then dopaquinone is produced. Apart from tyrosinase a, some proteins present in melanosomes, including tyrosinase-related protein types 1 and 2 (TYRP1, TYRP2), have key functions in eumelanin production. The role of TYRP1 is to increase the ratio of eumelanin: pheomelanin production, while it also plays its role as an anti-oxidant due to the presence of peroxidase activity [12,13]. A number of transcription factors, genes, and hormones including stem cell factor (SCF), SOX10, PAX3, WNT, MITF, ACTH, and α-MSH are involved in the melanin synthesis regulating pathway. It has been proposed that more than 125 genes contribute to its functional activity in melanogenesis. The defects in structural proteins, enzymes, ligands, hormones, and receptors of melanin pigment production result in loss of functions in these factors including *TYR*, *OCA2*, *TYRP1*, *TYRP2*, *SLC24A5*, *DCT*, *MATP*, and *MC1R*, which contribute to phenotypes of subtypes of OCA from OCA1–8 with an autosomal recessive non-syndromic pattern [13,14].

### 2.3. Prevalence/Epidemiology

Overall, the prevalence of OCA is projected at a frequency of 1 in 17,000–20,000. The mutant allele of OCA2 is carried in approximately 1 in 70 individuals, categorized as the prevalent type of albinism worldwide, although the frequency of mutation detection is higher in the OCA1 phenotype. The frequency of OCA2 alleles is more prevalent in the sub-Saharan African population, with a rate of 1 in 1000 cases [15,16]. This is due to the cultural customs that allow marriages within blood relations, a phenomenon is known as pseudo-dominance. In different populations, the prevalence of the specific types of albinism vary. The prevalence of type 1 of OCA (almost 1 in 40,000 worldwide) is the most common type (70% of cases) in America, China (sporadic albinism), the subcontinent (mostly familial albinism), and in the Caucasian population [17,18,19,20]. The frequency of OCA2 is the most common type with a prevalence rate of 1 in 39,000 worldwide. This type of albinism is more frequent in the population in sub-Saharan Africa, with an approximate rate of 1 in 3900, followed by African–Americans and Americans [15,16]. These two types of OCA are also seen in Pakistani and Indian families with frequent genetic mutations [9,17,21]. The prevalence of OCA type 3 and type 4 is 1 in 8500 and 1 in 100,000, respectively. OCA3 is reported in German, Japanese, and Indo–Pakistani populations. OCA4 is frequent in the Japanese population with a rate of 24% of overall OCA followed by different countries in Asia and Europe. OCA type 5 is only reported in a consanguineous Pakistani family [22], while cases of OCA6 and OCA7 are reported in China, eastern India, and Atlantic Island [23,24,25], but the exact prevalence has not been documented. Syndromic OCA is also variable in prevalence. The prevalence of Hermansky–Pudlak syndrome (HPS) is 1 in approximately 500,000 worldwide, the rate of Chediak–Higashi syndrome (CHS) is not evidently reported (almost 500 cases reported), and the prevalence rate for Angelman syndrome (AS) and Prader–Willi syndrome (PWS) is 1 in 12,000 and 1 in 15,000, respectively, i.e., a higher frequency rate than OCA types. The latter types of syndromic albinism are presented, such as OCA2, and 1% of cases showed gene deletions [26].

### 2.4. Phenotype Variations in OCA

Clinically, there are seven OCA forms that present a wide range of phenotypes, signs, and symptoms. OCA1 is the frequent type that is further divided into OCA1A and OCA1B subtypes [27,28,29,30]. OCA1A has a severe phenotype due to the complete absence of melanin pigment that leads to discoloration of skin and hair with a transparent iris in affected cases. All other forms of OCA including OCA1B and OCA2–OCA8 may present some coloration of the tissues such as hair, skin, and eyes with age development. The variety of pigmentation varies, displaying the extent of colors, but absolute pigment production is never achieved [31]. The photosensitivity of skin is also improved due to the absence of photo-protective melanin pigment in the skin. In OCA types, clinical presentations linked to eyes develop due to misrouting of the optic nerve and are presented as photophobia, congenital nystagmus, translucence hypo-pigmented iris, refractive errors, pigment scarcity in the epithelium, occasionally color vision impairment and foveal hypoplasia, reduction of visual acuity (VA) usually between 20/60 to 20/400, strabismus, and reduced stereoscopic vision due to misrouting of the optic nerve, which are the important OCA phenotype characteristics [32,33,34].

During the embryonic development stage, the melanin deficiency leads to eye developmental defects such as the retina and iris lacking melanin pigment, foveal hypoplasia, misrouting of optic nerve fibers of the visual cortex, and small or elongated globes [32]. As a result of melanin absence, a slight color is imparted to the eyes in albinism cases. Normal eye functioning is dependent on melanin pigment; therefore, impaired vision occurs due to low or deficient pigment availability. The main reason for impaired or loss of vision is under-development of the macula in the retina, leading to foveal hypoplasia. The other functions of melanin are to contribute to hair and skin coloration. The reduced production of melanin causes the development of a variety of colors in hair such as white to red or golden, light yellow or very light blonde [34]. In skin, the deficiency of pigment produces a milky-white and deep fair color in some cases. The complications of albinism result in severe damage of the skin from sunburn and may progress to skin cancers in later stages [34].

## 3. Clinical Features of Different OCA Types

Clinical characteristics of affected individuals are heterogeneous. OCA1A (MIM# 203100) is the subtype of OCA1, with white hair, eyebrows, eyelashes, and skin. There is no change in color throughout life. The iris color appears light-blue to pink and is absolutely transparent. The outcome of optic nerve misrouting is intense photophobia and nystagmus, and amelanotic nevi may be present. The cases with OCA1A show visual acuity 1/10 or even less, which persists throughout life without any difference between race and ethnicity [35].

Another subtype of OCA is OCA1B (MIM# 606952), which was previously known as yellow albinism. This type is similar to the OCA1A clinical characteristics, but there is scarce development of melanin pigment in the skin and hair. This pigmentation increases with age, and the iris color may be transformed to green/brown from blue color that is present at birth. Visual acuity is 2/10 in the OCA1B type. Another variant of OCA1 is temperature-sensitive OCA in which initially non-pigmented body hair is present but pigmented hair of hands and feet may develop with time due to temperatures difference between exposed and covered body parts [35,36].

OCA2-affected individuals may have some melanin pigmentation that imparts a wide range of skin, hair, and eye color; however, the color usually remains lighter in comparison to the relatives of the family member who are unaffected. Hair color is usually not completely white. The OCA2-affected individuals can develop benign lesions or dark spots due to extensive sun exposure and develop moles or pigmented nevi [15,16,37,38].

OCA3 is rufous albinism (red albinism of a rare type), and it was first documented in the African population. Individuals affected with OCA3 have yellow or reddish hair, red to brown skin, and brown eyes (contain excess melanin) or hazel eyes (green eyes due to less melanin). This type of OCA is reported mostly in Asian populations including Indian, Chinese, and Japanese, but is also found in Caucasian and northern European regions. In Asian origin, the color of affected individuals is fair or golden-haired, with light-brown eyebrows, and skin pigmentation is lighter compared to the skin color of the parents. In these affected albinos, the melanin pigmentation increases with age development. The degree of visual acuity impairment is less severe as examined in the affected cases of OCA1 and OCA2. Other important features of albinism including nystagmus and photophobia can be evident or may be absent [39,40,41].

Clinical features of OCA4 are consistent with the OCA2 features. In OCA4 cases, the range of pigmentation in the tissues varies from brown to yellowish brown. In these cases, the visual acuity can vary in severity from 20/30 to 20/400, which corresponds to the production of melanin and appears in tissue. Usually, the degree of visual acuity ranges between of 20/100 to 20/200 in OCA4 cases. OCA4 was initially documented in a case of Turkish origin but it has also expanded to other populations including German, Asian populations (Korean and Japanese), and subcontinent regions [25,42,43,44].

OCA5 type is only reported in affected individuals of Pakistan origin. These affected family members have white skin, golden hair, and ocular impairments as seen in the OCA1 type. The amount of visual acuity was 6/60 in one affected individual of a family [22].

OCA6 is characterized by hypopigmentation with white skin, brownish irises, and a range of hair color; golden, light brown to dark brown. Clinical features of OCA6 are not well determined because very few cases are investigated worldwide that cannot be differentiated based on specific features. The visual acuity of one affected individual measured 20/100, and fundoscopic analysis showed immature macula and revealed hypo-pigmentation. There was an absence of central fovea in the macula when compared to the brother in the same family by using optical coherence tomography [45]. In a recent study, clinical features documented in non-syndromic cases with OCA6 included photophobia, strabismus, nystagmus, and blue irises. The affected members had weak eyesight and they were not capable of seeing without using glasses [42].

The characteristics of clinical presentations of OCA7 include skin hypo-pigmentation, and the hair coloration varies from white-golden blond to dark brown hair in affected children from normal and carrier siblings or parents. The visual acuity was reduced in the affected individuals, ranging from 6/18 to 3/60, and other ocular problems associated with iris trans-illumination and nystagmus were also observed [46].

OCA8 is a recently investigated albinism type in Turkish and French ethnic groups. This type is a milder phenotype of albinism with moderate features including hypopigmentation of hair and skin, moderate foveal hypoplasia, nystagmus, and iris illumination [47]. On the other hand, the manifestations in the Turkish family were mild and presented with nystagmus, a small optic nerve, foveal hypoplasia, and mild iris trans-illumination [48].

## 4. Molecular Classification of Genes with Non-Syndromic OCA

Genetic studies in OCA congenital families described the mutation spectrum in genes that are causative of these anomalies. There are eight loci of OCA (OCA1–8) reported to date, and only seven genes are associated with the phenotypes, while one locus is still awaiting gene identification. Of these seven genes, the most commonly mutated genes are *TYR* and *OCA2*, and the summary of the function of OCA genes is outlined (Table 1). The detailed mutation spectrum for the causative genes associated with OCA are explained.

### 4.1. TYR (OCA1)

The most frequent albinism is OCA1 caused by a mutation in *TYR* (MIM# 606933). The physical localization of the *TYR* gene is present on chromosome 11q14.3, which has five coding exons. It is involved in the regulation of biosynthesis of the tyrosinase enzyme that mediates the rate-limiting chemical reactions in melanin formation. These crucial reactions are oxidation and hydroxylation of L-DOPA and DOPAquinone, the intermediate product [56]. *TYR* mutations due to defective tyrosinase activity lead to a diminished or absolute deficiency of melanin, with variable coloration in hair, skin, and eyes [57]. OCA1A (MIM# 203100) is a subtype of albinism in which a lack of tyrosinase results in the most severe clinical phenotype. The characteristics of OCA1A present with diminished skin, hair, and iris pigmentation linked to null or pathogenic alleles of *TYR*. On the other hand, OCA1B (MIM# 606952) is a subtype with positive tyrosinase activity that is a mild to moderate type of OCA outcomes due to less severe *TYR* mutations. This type of OCA presents reduced disease impact, as melanin pigmentation with less concentration is seen [9,20].

Clinical characteristics in temperature-sensitive OCA1 cases usually present white coloration of tissues including skin and hair, and blue eyes are present at birth. Although there is development in coloration in different parts of the body in adolescence, the visible parts of the body still carry the white hair [58]. This phenomenon is due to *TYR* mutation in temperature-sensitive amino acids that enhance the enzymatic activity, resulting in reduced or absent enzymatic activity which fails to execute the complete function. The mechanism of missense *TYR* mutations causes the retention of the TYR protein in the endoplasmic reticulum, and consequently lysosomal degradation develops loss of regulatory and synthetic functions of the gene [59]. Pathogenic *TYR* mutations are reported widely, and more than 400 variations are documented in this gene. In the Pakistani population, the large cohort of OCA families is linked to *TYR* mutations [21,49,60,61,62,63,64,65].

### 4.2. OCA2 (OCA2)

OCA2 is the second most investigated albinism type due to the defective *OCA2* gene (MIM# 203200) that was earlier reported as the *P* gene. *OCA2* is physically located on the chromosome at position 15q11.2–q12 and contains 23 coding exons with 345 kb nucleotide sequences. This gene mediates a 110 kDa protein that encodes 838 amino acids and has 12 transmembrane helices [66]. The P protein is a part of the Na+/H+ antiporter family and it contributes to regulating pH and tyrosinase activity for homeostasis in melanosomes [67]. Although the main functions of the OCA2 protein are to sort and transport tyrosinase and tyrosinase-related protein 1 (TYRP1) into the plasma membrane [68], it appears to take part in pH stabilization in the melanosome environment. Defects in the OCA2 gene result in the OCA2 type, and more than 400 mutations are reported in this gene [69]. The OCA2 alleles are the most frequent worldwide and are investigated in diverse populations including African, Caucasian, and Asian groups [52,70]. In the Pakistani population, several consanguineous families are reported with *OCA2* mutations, and it is categorized as the second most putative factor for OCA development [9,21,42,61,63,64].

### 4.3. TYRP1 (OCA3)

The third type of albinism, OCA3 (MIM 203290), is associated with *TYRP1* mutations (MIM# 115501). This gene spans a nucleotide sequence of 17 kb and is located on chromosome 9p23. It contains eight coding exons and encodes 536 amino acids that constitute tyrosinase-related protein-1 (Tyrp1) of ~61 kDa size. As TYRP1 has structural and conformation similarities to tyrosinase (GenBankNM_000550), it contributes to melanin synthesis by stabilizing tyrosinase and melanosome integrity and takes part in cellular proliferation and apoptosis to maintain the environment. It is also involved in the oxidation of 5,6-dihydroxyindole-2-carboxylic acid (DHICA), a step in the melanin biosynthetic pathway. Earlier studies reported OCA3-associated cases in the African population only; nucleotide variations in *TYRP1* have also been investigated as follows: Caucasian origin, Asian–Indian populations, and Chinese races [22,31,42,61,64,71].

### 4.4. SLC45A2 (OCA4)

The OCA type 4 (OCA4, OMIM 606574) results due to causative mutations in the *SLC45A2* gene (earlier, it was reported as a membrane-associated transporter protein (*MATP*) gene). It is localized on chromosome 5p13, spanning a nucleotide size of 40 kb and having seven coding exons [72]. It regulates the production of 530 amino acids containing a membrane-bound transporter protein, SLC45A2, with a size of ~58 kDa, with 12 transmembrane helices. This protein modulates the role in the proper binding of copper to tyrosinase and maintenance of melanosomal pH necessary for normal tyrosinase activity [46]. The primary cases of *MATP*/*SLC45A2* mutations were identified in Turkish ancestry, and then the reports were expanded to other populations such as the Chinese, German, Japanese, Indo–Pak, and Korean ethnic groups. This type of OCA is commonly found in the Japanese population; however, *SLC45A2* mutations are rare in Pakistani OCA patients [22,52,55].

### 4.5. OCA5

The OCA5 locus on chromosome 4 (4q24) consists of fourteen genes, but the particular causative gene of OCA5 has not yet been identified. Kausar et al., 2013 reported OCA5 in a single Pakistani family [22].

### 4.6. SLC24A5 (OCA6)

The OCA6 type of albinism occurs due to *SLC24A5* (solute carrier family-24, member 5; OMIM: 609802) mutations, and the gene is localized on chromosome 15q21.1 [45]. This gene mediates the regulation of the SLC24A5 protein that is a constituent of potassium-dependent sodium/calcium exchanger-5 (NCKX5) and it contributes to melanosome maturation and is involved in the process of melanin production [23]. *SLC24A5* mutations were earlier identified in the Chinese population, but reports from various ethnic origins are documented that present the heterogeneity of OCA6. The manifestation of the cutaneous phenotype was different to hair color ranging from white to blond or dark brown. The SLC24A5 protein existence in melanosomes play a vital role in melanosome structure and synthesis. Therefore, a defective gene or truncated SLC24A5 protein may negatively impact the development of melanosomes and the production of melanin [23,24,25,73,74]. In the Pakistani population, very few cases of albinism are reported causative of *SLC24A5* mutations. In Iranian families, mutations are identified as causative of OCA6 [70].

### 4.7. C10ORF11 (OCA7)

The OCA7 (MIM: 615179) type of albinism is associated with *C10orf11* mutations, and this gene is physically localized on chromosome 10 at the 10q22.2–q22.3 position. *C10orf11* mediates the synthesis of a Leucine-Rich Melanocyte Differentiation-Associated Protein, consisting of 226 amino acids with one LRR C-terminal (LRRCT) and three domains of leucine-rich repeats (LRRs). The C10orf11 protein plays a role in the melanin pathway including cell-adhesion, differentiation of melanocyte differentiation, assembly of the extracellular matrix, cell-signaling, RNA processing, and neuronal development [46,75]. The genetic mutations of *C10orf11* associated with OCA7 have been identified in Indian and Iranian populations [64,76].

### 4.8. DCT Associated with OCA8 (New Type)

A new type of OCA is OCA8 (MIM 191275) linked to *DCT* mutations investigated recently. The gene is localized on chromosome 13q31–q32 [77]. The dopachrome-tautomerase (*DCT*) gene, also called *TYRP2*, has eight coding exons and it mediates the TYRP2 protein that has 80% similarity to TYRP1. The DCT protein along with TYRP1 and TYR plays an important role in melanin biosynthesis and melanosome apoptosis [78]. Recently, the causative mutations in *DCT* have been detected in the French and Turkish populations [47,48].

## 5. Mutation Spectrum of OCA Genes in the Pakistani Population

Pakistan is situated in the subcontinent with a historical cross-side associated with the various political and cultural impeders and had been a bridge for many invaders and reigns in South Asia. Historical migrations in this region have ensued in a combination of antique cultures, which continue the social or cultural blend of this population. Inbred and close familial relationships make Pakistan a productive country to conduct genetic differentiation studies and explore the various genomic characteristics of a human population [79]. Genetic diseases have been widely investigated in the diverse and inbred population of Pakistan. This area of study markedly enhanced genetic facilities in the society and encouraged researchers to play a further role in this category. The Pakistani population consists of variable language and ethnic groups in which the customs values are considered at the top. This concept progressively develops in marriages with close relations and aggravates the consanguinity. This consanguinity results in the genetic disorders that are impeding the population.

OCA is widely investigated and linked to different ethnic or language groups including the Kashmiris of Azad-Jammu Kashmir (federally controlled by Pakistan) and the Punjabi, Sindhi, Balochi, Pashtoon, and Urdu speaking language groups. Different investigators detected genetic mutations in OCA-associated genes, predominantly *TYR* and *OCA2*. More than ninety consanguineous OCA families have been explored to determine the *TYR* mutations [21,42,49,61,63,64,65]. In the *TYR* gene, a variety of novel, known, and founder mutations have been demonstrated by various research reports. In the population, the most frequent and common mutations of the *TYR* gene are c.832C>T and c.1255G>A; p.Gly419Arg that are reported in 21 families (22.6%) and 20 families (23.8%), respectively. On the other hand, some mutations are founders that contribute repeatedly to the ancestry i.e., c.230G>A (5.9%). In Pakistani consanguineous families, about 90 mutations are associated with *TYR*-linked OCA that includes missense 86.9%, nonsense 7.1%, and the remaining splice site error, deletion, and frameshift mutations (Table 2).

The second gene most widely investigated in Pakistani OCA families is *OCA2*. This gene is linked to more than fifty OCA families with variable mutation patterns (Table 3). From the reported data, about 17 families are linked to splice-site error c.1045–15 T>G that accounts for 30.3% of total mutations in *OCA2* [9,21,64]. In a study from Pakistan, it was observed that this mutation was present in a homozygous state in some of the affected individuals, compared with heterozygous form in other individuals with variable OCA phenotypes among these affected family members. Interestingly, this variant presents the mechanism of missing heritability, which is linked to almost 25% patients being homozygous for OCA alleles [57]. It has been described that the presentations of OCA2 can be modified by the genes of pigment pathways such as TYRP1 or MCIR, which has synergistic activity [80].

There are very few mutations reported in other genes including *TYRP1*, *SLC45A2*, and *SLC24A5* in Pakistani families (Figure 2), with frequencies of 5.9%, 3.5%, and 1.9%, respectively [21,42,64].

**Table 2 genes-13-01072-t002:** List of *TYR* gene mutations reported in OCA families from Pakistan.

Mutation	Language/Ethnic Group	Mutation Type and Status	No. Pakistani Families Linked to OCA	References
c.62 C>T; p.Pro21Leu	Punjabi	Missense/homozygous	1	[49]
c.103 T>C; p.Cys35Arg	Punjabi	Missense/homozygous	2	
c.132T>A; p.Ser44Arg	Kashmiri, Pakhtoon, Baluchi	Missense/homozygous	3	[9,81]
c.164G>C; p.Cys55Ser	Punjabi	Missense/homozygous	1	[64]
c.223G>T; p.Asp75Tyr	Punjabi	Missense/homozygous	1	
c.230G>A; p.Arg77Gln	Kashmiri	Missense/homozygous	4	[63]
c.248T G; p.Val83Gly	Pakhtoon	Missense/homozygous	1	[61]
c.240G>C; p.Trp80Cys	Pakhtoon	Missense/homozygous	2	[9]
c.272G>A; p.Cys91Tyr	Punjabi	Missense/homozygous	1	[65]
c.308G>A; p.Cys103Tyr	Punjabi	Missense/homozygous	1	
c.346C>T; p.Arg116 *	Saraiki, Punjabi	Nonsense/homozygous	2	[60,65]
c.575C>A; p.Ser192Tyr	Kashmiri	Missense/homozygous	1	
c.585G>A; p.Trp195 *	Punjabi	Nonsense/homozygous	1	[64]
c.593 T>C; p.Ile198Thr	Pakistani	Missense/homozygous	1	[62]
c.715C>T; p.Arg239Trp	Kashmiri	Missense/homozygous	1	[63]
c.832C>T; p.Arg278 *	Urdu speaking, Punjabi, Pakhtoon	Nonsense/homozygous	21	[9,21,42,49,63,64,65]
c.826T>C; p.Cys276Arg	Pakhtoon	Missense/homozygous	1	[42]
c.895C>T; p.Arg299Cys	Saraiki	Missense/homozygous	1	[61]
c.896A>G; p.Arg299His	Punjabi, Pakhtoon	Missense/homozygous	3	[56,64]
c.943–948delTCAGCT; p.315–316delSerAla	Punjabi	Deletion/heterozygous	1	[64]
c.982G>C; p.Glu328Gln	Pakistani origin (language/ethnicity not documented)	Missense/homozygous	1	[36]
c.1037 G>A; p.Gly346Glu	Punjabi	Missense/homozygous	1	[64]
c.1037G>T; p.Gly346Val	Punjabi	Missense/homozygous	1	
c.1037–7 T>A	Punjabi	Splicing error/heterozygous	4	
c.1037–18 T>G	Urdu speaking	Splicing error/heterozygous	1	
c.1147 G>A; p.Asp383Asn	Pakhtoon	Missense/homozygous	1	
c.115 del 340-341;frame shift p.R278X	Pakistani	Fame shift nonsense	1	[59]
c.1184 + 2 T>C	Punjabi	Splicing error/heterozygous	1	[64]
c.1204 C>T; p.Arg402 *	Punjabi	Nonsense/homozygous	1	
c.1217 C>T; p.Pro406Leu	Punjabi	Missense/homozygous	1	[64]
c.1231 T>C; p.Tyr411His	Punjabi	Missense/homozygous	1	
c.1255G>A; p.Gly419Arg	Punjabi, Pakhtoon, Sindhi, Kashmiri, Saraiki	Missense/homozygous	23	[9,21,49,60,61,63,64,65,81]
c.1424G>A; p.Trp475 *	Saraiki	Nonsense/homozygous	1	[61]
Exons 4–5 deletion	Punjabi	Deletion/heterozygous	2	[64]
**Total families linked to *TYR***	**Ninety**			

* is presentation of non-sense or truncated mutation.

**Table 3 genes-13-01072-t003:** *OCA2* gene mutations in OCA families from Pakistan.

Mutation	Language/Ethnic Group	Mutation Type and Status	No. Pakistani Families Linked to OCA	References
c.827T>A; p.Val276Glu; c.877G>C; p.Glu293Gln	Kashmiri	Missense/compound heterozygous	1	[21]
*TYR*-c.832C>T; p.Arg278 *; OCA2-c.954G>A; p.Met318Ile	Sindhi	Missense/non-sense digenic/compound heterozygous	1	
TYR-c.649C>T; p.Arg217Trp; *OCA2*-c.1456G>T; p.Asp486Tyr	Sindhi	Missense digenic/compound heterozygous	1	
*TYR*-c.1255G>A; p.Gly419Arg; *OCA2*-c.954G>A; p.Met318Ile	Punjabi	Missense digenic/compound heterozygous	1	
c.2079 + 5G>T	Pakhtoon	Splice site/compound heterozygous	1	[40]
Ex19 del	Pakhtoon	Frame-shift/homozygous	2	[61]
c.1327G>A; p.Val443Ile; c.1762C>T; p.Arg588Trp	Punjabi	Missense/compound heterozygous	1	[9]
Exons 3–14 deletion	Punjabi	Frame-shift/homozygous	1	[64]
Exons 7–8 deletion	Punjabi	Frame-shift/homozygous	2	
c.1045–15 T>G	Punjabi, Saraiki, Sindhi	Splice site/compound heterozygous	17	[9,49,64]
c.1056 A>C	Sindhi	Missense/homozygous	1	[64]
c.1064 C>T	Sindhi	Missense/homozygous	1	
c.1075 G>C	Saraiki	Missense/homozygous	1	
c.1182 + 2 T>TT	Punjabi	Splice site/compound heterozygous	1	
c.1211 C>T	Punjabi	Missense/homozygous	1	
c.1322 A>G	Sindhi	Missense/homozygous	1	
c.1456 G>T	Punjabi, Saraiki, Sindhi	Missense/homozygous	11	[49,64]
c.1922 C>T	Punjabi	Missense/homozygous	1	[64]
c.1951 + 4 A>G	Urdu speaking	Splice site/compound heterozygous	1	
Exon 19 deletion	Punjabi	Deletion/compound heterozygous	1	
Exons 20–24 deletion	Pakhtoon, Punjabi	Deletion/compound heterozygous	2	
c.2020C>G; p.Leu674Val.; c.408_409delTT; p.Arg137Ilefs * 83	Pakhtoon	Missense/frame shift/compound heterozygous	1	[9]
c.2228 C>T; p.Pro743Leu	Urdu speaking, Punjabi	Missense/homozygous	2	[49,64]
c.954 G>A; p.Met318Ile	Punjabi	Missense/homozygous	1	[49]
c.1580 T>G; p.Leu527Arg	Punjabi	Missense/homozygous	1	
c.2359 G>A; p.Ala787Thr	Punjabi	Missense/homozygous	1	
c.2360 C>A; p.Thr787Arg	Punjabi	Missense/homozygous	1	
c.2360 C>T; p.Thr787Ile	Punjabi	Missense/homozygous	1	
c.2458T>C; p.Ser820Pro	Punjabi	Missense/homozygous	1	[9]
**Total families linked to *OCA2***	**Fifty nine**			

* is presentation of non-sense or truncated mutation.

## 6. Conclusions

Autosomal recessive non-syndromic oculocutaneous albinism (OCA) is increasingly devastating in populations with variable languages or ethnic groups, having customs of consanguineous marriages, including in Pakistan. These congenital or familial genetic disorders have augmented the health and social issues in the communities. OCA is a rare genetic anomaly with diverse heterogeneity and its clinical characteristics are evident in skin, eyes, and hair. The defects in enzymes that regulate melanin biosynthesis in melanosomes are structurally or functionally altered, and the outcome is various types of OCA. Molecular studies identified eight different loci of OCA in which seven OCA phenotypes were linked to putative gene mutations including *TYR*, *OCA2*, *TYRP1*, *SLC45A2*, *SLC24A5*, *C10orf11*, and *DCT*. The mutations linked to these genes are reported worldwide. The OCA5 locus has also been documented for the OCA phenotype, but the gene has not been identified. In the Pakistani population, more than two hundred families have been reported to have different syndromic and non-syndromic OCA gene mutations. The most frequent gene mutated in this population is *TYR* followed by *OCA2*. The mutations in *TYRP1*, *SLC45A2*, and *SLC24A5* genes are reported in the Pakistani population but the frequency is very low. A large cohort of families is still not linked to any known gene/s of OCA, which may predict the possibility of mapping new genes or events for the development of OCA disease. Next-generation sequencing (NGS) techniques are versatile tools to explore the hidden molecular factors associated with albinism. Using these state-of-the-art technologies might improve the diagnostic prospective and clinical understanding of the disorders. The establishment of NGS in developing countries such as Pakistan would improve the diagnostic facilities that might have an impact on disease-controlling factors. Updating the data about OCA will be helpful for researchers and clinicians to learn more about the molecular diagnostic facilities and clinical phenotypes that can improve counselling services for the families and their tailored management. Further, it might help the research to explore new ideas to determine the role of genes in therapeutic medicine. The study is limited with respect to discussing the genotype and phenotype outcome of the mutations in respective gene/s that further require comprehensive data to depict the role of these genes in the disease.

## Figures and Tables

**Figure 1 genes-13-01072-f001:**
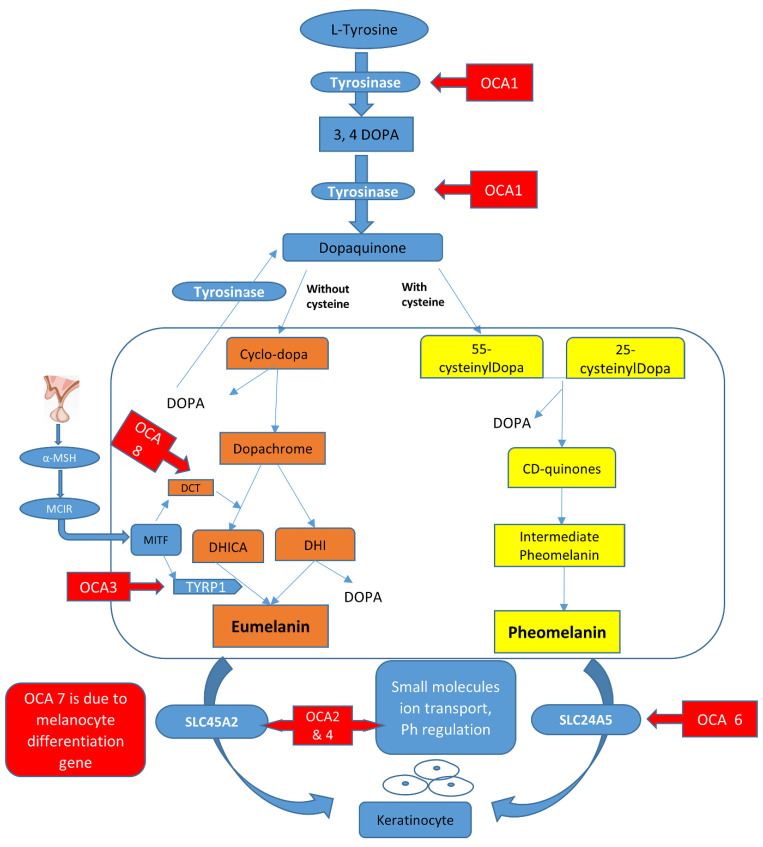
Graphical representation of the melanin synthesis pathway and rate-limiting steps that if defective lead to OCA types.

**Figure 2 genes-13-01072-f002:**
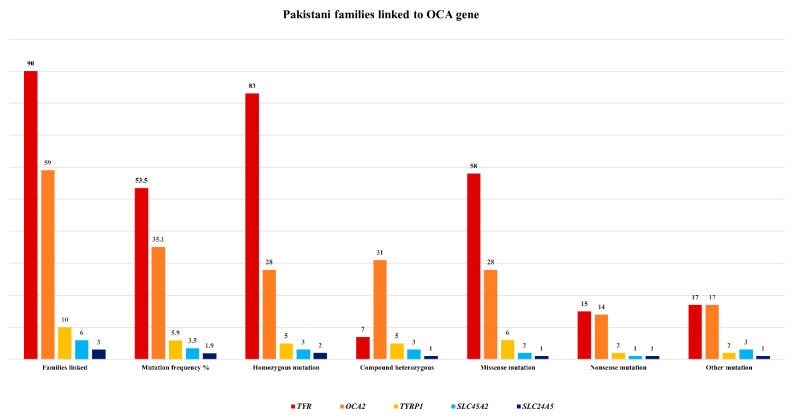
Pakistani families linked to different OCA genes with frequency of mutations.

**Table 1 genes-13-01072-t001:** Description of known genes and their cellular functions and localization in oculocutaneous albinism.

Locus Name	Physical Location	Causative Gene	Cellular Location	Proposed Function	Reported Mutations	References
**OCA1**	11q14	** *TYR* **	**Melanosome**	Malanogenesis	487	[42,49,50]
**OCA2**	15q11	** *OCA2* **	**Transmembrane of melanosome**	Regulate and transport tyrosinase	364	[51]
**OCA3**	9p23	** *TYRP1* **	**Melanosome**	Stabilize tyrosinase	66	[52,53,54]
**OCA4**	5p13	** *SLC45A2* **	**Transmembrane of melanosome**	Maintain melanosomal pH and help in binding of copper to tyrosinase	86	[40,55]
**OCA5**	4q24	** *-* **	**-**	-	-	[22]
**OCA6**	15q21	** *SLC24A5* **	**Melanosome**	Ion exchange in melanocytes	37	[22,45]
**OCA7**	10q22	** *C10orf11* **	**Melanocyte**	Melanocyte Differentiation	6	[46]
**OCA8**	13q31	** *DCT* **	**Melanocyte**	Melanosome apoptosis	4	[47,48]

## Data Availability

Not applicable.

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
