# Peer review of "Clinical and Mutation Spectrum of Autosomal Recessive Non-Syndromic Oculocutaneous Albinism (nsOCA) in Pakistan: A Review"

_genes, 2022, doi:10.3390/genes13061072_

Round 1

Reviewer 1 Report

Thanks to the author for contributing to a rare Genetic disease, followings are my comments/concern to improve the manuscript:

1.     Since the current review only describes a country (Pakistan) population. I feel the title of the manuscript should be revised.

2.     References are too old and 50 references are sufficient for the review study.

3.     Statement, in conclusion, is not very much clear (first few lines).

4.     Please describe the limitations of the study.

5.     Punctuations are required throughout the manuscript.

6.     Line number 15: The complications of OCA develop skin cancer and variable syndromes (what are those variable syndromes).

7.     Line number 14-18: No need to mention cousin marriage here

8.      In line 28, rephrasing is required.

9.      In the line number 43, some words are missing

10.                         Under the heading 2.2, only three references are mentioned, please give credit to regional workers

11.                         In line 105, no need to explain pseudo-dominance.

12.                        Line number 343, OCA is already explained as Oculocutaneous albinism. No need to write the full name again.

13.                         Line 374, It must be Punjabi rather than Punjab.

14.                         References are not having the uniformity

Author Response

Point 1:   Since the current review only describes a country (Pakistan) population. I feel the title of the manuscript should be revised.

Response 2: Title has been revised accordingly.

Point 2: References are too old and 50 references are sufficient for the review study.

Response 2: The references cover all the mutations which are reported to a specific gene, therefore impact of each mutation is required.

According to the journal (Genes) guidelines, for review, the references are given in this manuscript.

Point 3: Statement, in conclusion, is not very much clear (first few lines).

Response 3: The statements in the conclusion have been revised for a more understanding of the ideas.

Point 4: Please describe the limitations of the study.

Response 4: Limitation to the study is written.

 Point 5: Punctuations are required throughout the manuscript.

Response 5: The punctuations are written in throughout the document where required.

 Point 6: Line number 15: The complications of OCA develop skin cancer and variable syndromes (what are those variable syndromes).

Response 6: The example of OCA associated syndromes has been mentioned in the abstract.

 Point 7: Line number 14-18: No need to mention cousin marriage here

Response 7: The suggested word is deleted.

Point 8: In line 28, rephrasing is required.

Response 8: The sentences are rephrased accordingly.

 Point 9: In the line number 43, some words are missing

Response 9: The changes have been made and statements are clarified.

 Point 10: Under the heading 2.2, only three references are mentioned, please give credit to regional workers

Response 10: The basics of albinism are described here comprehensively while all the workers who gave their input have been cited at the different places accordingly.

Point 11: In line 105, no need to explain pseudo-dominance.

Response 11: The explanation is removed.

Point 12: Line number 343, OCA is already explained as Oculocutaneous albinism. No need to write the full name again.

Response 12: Te abbreviation of OCA is written and the detailed name is removed.

Point 13: Line 374, It must be Punjabi rather than Punjab.

Response 13: The wording is changed accordingly.

Point 14: References are not having the uniformity

Response 14: All the references in the bibliography is revised according to the formatting style of the journal.

Reviewer 2 Report

The update about OCA in the form of a review article is much needed and may be a good piece of knowledge for the readers, however, this manuscript by Kalimullah et al needs thorough revision to present it in a more clear, organized and easily understandable format. Following are a few suggestions.

The title is comprehensive but the content is focused on a single population, it will be of interest if authors compare the prevalence and clinical presentation of OCA among different populations including Pakistan.

The paragraph in the introduction, regarding syndromic and non-syndromic albinism (Line 37-38) is not apprehendable.  Authors need to rewrite it and describe the difference between the two types of diseases clearly with basic information about the genes of syndromic albinism.

As this is a review article, the authors should describe the methodology for the search of the literature in the Methods section, but they are describing the results directly in Material and Method section.

Authors should be consistent about subtypes/clinical types of OCA, either OCA 1 to7 or OCA1 to 8. Please refer to Line 67 and line 92.

It will be more beneficial and interesting for readers if authors describe each gene/locus of OCA with a different main heading and then elaborate on the epidemiology, prevalence, phenotype, clinical and genetic variation as subheadings.

The language also needs revision for improvement. In addition, the authors should be consistent with the explanation of phenotypic terms such as lines 29-30 ‘’ two forms of albinism categorized as ocular (only pertaining to eyes) and oculocutaneous albinism.”

Author Response

Point 1: The title is comprehensive but the content is focused on a single population, it will be of interest if authors compare the prevalence and clinical presentation of OCA among different populations including Pakistan. 

Response 1: The review is focusing mainly on the cases of familial albinism, therefore the data available on congenital disease is more prominent in the Pakistani population. The data was searched from the other ethnic groups but very few familial OCA cases are reported. Mostly the cases of OCA in different populations are sporadic. The main focus of this review was to describe the data related the familial disease.

The title is revised by specifying the population.

Point 2: The paragraph in the introduction, regarding syndromic and non-syndromic albinism (Line 37-38) is not apprehendable. Authors need to rewrite it and describe the difference between the two types of diseases clearly with basic information about the genes of syndromic albinism.

Response 2: Brief description of non-syndromic and syndromic albinism is provided. More details are mentioned in the literature review part of the review article.

Point 3: As this is a review article, the authors should describe the methodology for the search of the literature in the Methods section, but they are describing the results directly in the Material and Method section.

Response 3: This is a general review in which the previously published data on the relevant issue is searched from the different websites to document the results of previous studies. Therefore, there is no materials and methods or result description. However, a general comment can be given here about the data recruiting. All the relevant published data were retrieved from PubMed, Web of Science, Scopus, and Google Scholar.

Point 4: Authors should be consistent about subtypes/clinical types of OCA, either OCA 1 to7 or OCA1 to 8. Please refer to Line 67 and line 92.

Response 4: OCA types are made consistent according to the recommendations.

Point 5: It will be more beneficial and interesting for readers if authors describe each gene/locus of OCA with a different main heading and then elaborate on the epidemiology, prevalence, phenotype, clinical and genetic variation as subheadings.

Response 5: The reviewer's suggestion is remarkable but this will create disorganization of the data to comprehend the features of different clinical types. Therefore, the clinical, epidemiological and mutation spectrum are discussed comprehensively, so that readers can gain the information specifically.

Point 6: The language also needs revision for improvement. In addition, the authors should be consistent with the explanation of phenotypic terms such as lines 29-30 ‘’ two forms of albinism categorized as ocular (only pertaining to eyes) and oculocutaneous albinism.”

Response 6: Te language has been improved according to the suggestion in throughout the document.

There are two main forms of albinism; ocular and oculocutaneous. The phenotypes are different in both categories. In ocular albinism, the phenotypes are associated with eyes only while in oculocuatenous albinism the phenotypes are linked to eyes, hair and skin. The statement is revised in the text.

Reviewer 3 Report

A comprehensive literature review is represented in this article. Author have sufficient knowledge to cover all aspects, which fall under the main heading. However some minor errors are observed which are as follows

Typos, format and grammatical errors are seen through out the text which need revision.

Material and methods heading is inappropriate in review article context without any original experimental work and results. Most of the data represented under this heading is representing data from the literature.

Ratios represented for the prevalence of the disease need more clarification. (for worldwide or specific population)

Kindly revisit molecular classification of genes with Non-Syndromic OCA section to avoid any plagiarism.

In conclusion section, future perspectives of this review article needs more elaboration. 

Author Response

Point 1: Typos, format and grammatical errors are seen throughout the text which need revision.

Response 1: The language errors are screened and amended throughout the document. The changes are tracked.

Point 2: Material and methods heading is inappropriate in review article context without any original experimental work and results. Most of the data represented under this heading is representing data from the literature.

Response 2: This is a review article therefore material methods is not required. The wording is replaced with a literature review.

Point 3: Ratios represented for the prevalence of the disease need more clarification. (for worldwide or specific population).

Response 3: The title of the review is revised to focus on the specific population. Actually, this review highlights more about the familial type of OCA which is commonly reported in the Pakistani population. Also, the published data is more pronounced in the consanguineous families in this population whereas sporadic OCA is present in variable ethnic groups.

Point 4: Kindly revisit molecular classification of genes with Non-Syndromic OCA section to avoid any plagiarism.

Response 4: This section is carefully written and the plagiarism was analyzed before submission to the journal. The plagiarism data is fulfilling the guidelines of the journal and it is not exceeding overall and in this section.

The reviewer may have access to the plagiarism link here;

Plagiarism file of OCA review

Point 5: In the conclusion section, future perspectives of this review article need more elaboration.

Response 5: Conclusion is improved and more perspectives are mentioned according to the reviewer’s recommendation.

Round 2

Reviewer 2 Report

minor grammatical errors need to be checked.